# Site-specific electrodeposition enables self-terminating growth of atomically dispersed metal catalysts

Yi Shi[1], Wen-Mao Huang [1], Jian Li[1], Yue Zhou[1], Zhong-Qiu Li[1], Yun-Chao Yin[1] & Xing-Hua Xia [1✉]

The growth of atomically dispersed metal catalysts (ADMCs) remains a great challenge owing to the thermodynamically driven atom aggregation. Here we report a surface-limited electrodeposition technique that uses site-specific substrates for the rapid and room-temperature synthesis of ADMCs. We obtained ADMCs by the underpotential deposition of a non-noble single-atom metal onto the chalcogen atoms of transition metal dichalcogenides and subsequent galvanic displacement with a more-noble single-atom metal. The site-specific electrodeposition enables the formation of energetically favorable metal–support bonds, and then automatically terminates the sequential formation of metallic bonding. The self-terminating effect restricts the metal deposition to the atomic scale. The modulated ADMCs exhibit remarkable activity and stability in the hydrogen evolution reaction compared to state-of-the-art single-atom electrocatalysts. We demonstrate that this methodology could be extended to the synthesis of a variety of ADMCs (Pt, Pd, Rh, Cu, Pb, Bi, and Sn), showing its general scope for functional ADMCs manufacturing in heterogeneous catalysis.

[1] State Key Laboratory of Analytical Chemistry for Life Science, School of Chemistry and Chemical Engineering, Nanjing University, Nanjing 210023, P.R. China. ✉email: xhxia@nju.edu.cn

Atomically dispersed metal catalysts (ADMCs), with maximum atom efficiency and unique metal coordination environments, are considered as an important frontier in heterogeneous catalysis research[1–3]. However, single-metal atoms tend to form aggregates driven by the decrease of surface free energy[4,5]. In recent years, many methods based on physical and chemical approaches have emerged for the synthesis of ADMCs[6,7]. Most of these methods either depend on expensive equipment and sophisticated techniques, or suffer from the harsh conditions used (high temperatures/pressures), low metal loading, and time-consuming procedures. A universal strategy for fast, mild, scalable and controllable ADMCs synthesis remains in high demand.

We considered whether we could synthesize ADMCs by simply using the room-temperature electrodeposition (ED) method. Owing to its facile operation, mildness, large scalability, low cost, and precision[8], ED has been extensively used for the fabrication of nanomaterials and industrial applications (e.g., electroplating)[9,10]. However, this process leads to the inevitable formation of multilayer bulk structures with nonuniform coverage (Fig. 1(i) and Supplementary Fig. 1a). To circumvent this difficulty, surface-limited techniques—typically underpotential deposition (UPD), a process by which monolayers form at potentials above the thermodynamic value—have been used to afford single-layer coverage (Fig. 1(ii) and Supplementary Fig. 1b)[11]. Examples of the successful use of this method include the syntheses of a two-dimensional (2D) Pt layer on a Au electrode by the quenching effect of underpotential-deposited hydrogen by the Moffat group[12]

and of various single-layer core-shell metallic nanostructures by the Adzic group[13,14]. Owing to the uniform distribution of the interfacial work function of the supporting substrate, which is composed of the same (homo-) element, only limited success has been achieved in single-atom synthesis.

We reasoned that the single-atom layer could potentially be upgraded to single atoms deposited site-specifically on a supporting substrate for UPD (Fig. 1(iii) and Supplementary Fig. 1c) if the substrate consists of isolated active sites owing to the nonuniform work function (Supplementary Note 1). Transition metal dichalcogenides (TMDs) possessing isolated and nonuniformly distributed chalcogen sites (e.g., S, Se, and Te)—elements that have Lewis base character with lone pair electrons and suitable electronegativity[15]—can potentially interact with, and hence stabilize single-metal atoms. Owing to the strong coordination and electronic interaction between the depositing metal and the chalcogen sites on the substrate, the metal–support interaction is energetically favorable compared to the metal–metal interaction in the crystal lattice of the bare metal during the UPD process. After the formation of a metal–support bond, the sequential deposition of metal–metal bonds is forbidden at the UPD potential (Fig. 1(iii)). Growth from a single atom would be self-terminated and would be automatically and strictly controlled. Site-specific UPD provides the activation energy and fast kinetics for the formation of thermodynamically favored metal–support bonds and atomic dispersion. The self-termination enables the synthesis of single atoms preferentially constrained at these specific sites under the UPD process on a

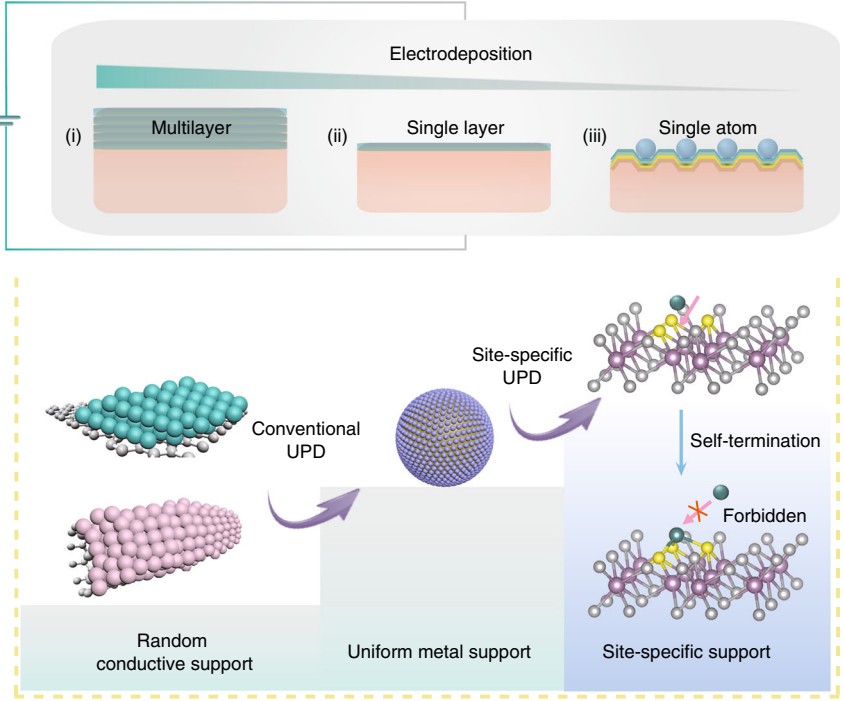

**Fig. 1 Methodological development of ADMC preparation by electrodeposition.** The blue wedge structure represents the downsizing of electrodeposition on nanomaterials from a multilayer (i) to a single layer (ii) and further to single atoms (iii). Representative examples of each electrochemical method are shown: (i) Electrically conductive substrates with high surface areas (e.g., graphene or carbon nanotubes) are often used to electrodeposit metal nanoparticles or metallic thin films. However, this process typically leads to the formation of a multilayered bulk phase (represented by the blue or pink ball constructed nanostructures in the figure). (ii) Conventional underpotential deposition (UPD), typically of a metal cation (e.g., $Cu^{2+}$, $Ag^+$, $Bi^{3+}$, or $Pb^{2+}$) deposited onto uniform metal support (e.g., Au, Pd, or Pt) at a potential more positive than its equilibrium potential (the potential at which it deposits onto itself), has been widely investigated for the growth of a 2D single layer. (iii) Inspired by these findings, we proposed that the synthesis of ADMCs might be realized through site-specific electrodeposition on transition metal dichalcogenides (e.g., $MoS_2$) with isolated active sites for UPD. The site-specific UPD enables the energetically favorable deposition of single-atom metal on the chalcogen sites, and then automatically terminates the sequential aggregation of metal atoms.

timescale of minutes. This site-specific ED (SSED) method represents a major breakthrough with respect to other extremely time-consuming and procedurally demanding strategies.

## Results

**Synthesis and structural characterization of ADMCs**. We approached the fabrication of ADMCs by first exploring the UPD of Cu atoms on chemically exfoliated molybdenum disulfide (ce-MoS$_2$), which was used as a prototypical TMD substrate to demonstrate the feasibility of a site-specific UPD strategy for fabricating ADMCs. Chemical exfoliation by lithium intercalation (Supplementary Fig. 2) gives rise to a deformed crystal phase with disparate electronic properties in the ultrathin nanosheets of MoS$_2$ (Supplementary Fig. 3)[16–18]. We chose to utilize the metallic ce-MoS$_2$ material for its high conductivity, which potentially realized the charge transfer at the interface between MoS$_2$ support and electrode during ED process. The decreased deposition potential between two notable cathodic peaks (Supplementary Fig. 4) demonstrates the capability of ce-MoS$_2$ to deposit Cu at potentials more positive than the equilibrium potential of $E_{Cu^{2+}/Cu}$ (Supplementary Table 1). During the UPD of Cu atoms on ce-MoS$_2$, the cathodic current signal decreases rapidly and reaches a saturated plateau, where the growth of Cu atoms on the surface terminates automatically (Supplementary Fig. 4f, product termed Cu-SAs/MoS$_2$). No obvious nanoparticles or clusters were discernable in Cu-SAs/MoS$_2$, whereas elemental Cu was detected by X-ray photoelectron spectroscopy (XPS; Supplementary Fig. 4g–i), indicating the successful growth of a single atom on MoS$_2$ by using UPD (Supplementary Note 2)[19].

To deposit single-atom Pt on ce-MoS$_2$, PtCl$_4^{2-}$ ions were introduced into the Cu-SAs/MoS$_2$ system for the galvanic exchange of Cu atoms by Pt(II) (Supplementary Fig. 5). The open-circuit potential was measured to monitor the potential response during the replacement step (Cu$_{UPD}$ + Pt$^{2+}$ → Cu$^{2+}$ + Pt, Supplementary Fig. 5). Under these conditions, Pt was atomically dispersed on the ce-MoS$_2$ nanosheets (Pt-SAs/MoS$_2$). After functionalization, the zeta potential of the nanosheet suspensions significantly increased (Supplementary Table 2) owing to suppression of excess charges and the attachment of Pt species. The loading of Pt in Pt-SAs/MoS$_2$ was 5.1 wt%, as measured by inductively coupled plasma optical emission spectrometry (Supplementary Note 3), much higher than those in previously reported ADMCs (Supplementary Table 3). No Pt-containing clusters or nanoparticles were observed by conventional transmission electron microscopy (TEM) imaging (Fig. 2a). X-ray diffraction (XRD) revealed no crystalline Pt phases on the surface of the ce-MoS$_2$ nanosheets (inset of Fig. 2a). Electrochemical cyclic voltammogram did not display peaks characteristics of Pt in the regions of Pt-H adsorption/desorption, indicative of the discrete distribution of Pt (Supplementary Fig. 6). It was confirmed by aberration-corrected high-angle annular dark-field-scanning TEM (HAADF-STEM) that the Pt atoms (the brightest white dots in Fig. 2b) were atomically dispersed on ce-MoS$_2$. A magnified HAADF-STEM image (Fig. 2c) and its corresponding density functional theory (DFT)-optimized structural model (Fig. 2d and Supplementary Fig. 6, and Supplementary Table 4) provided evidence for Pt attachment on the Mo top site (Fig. 2e). A slight discrepancy could be explained by the localized structural distortion due to the attachment of Pt atoms[20,21]. Moreover, analysis by STEM coupled with energy-dispersive X-ray spectroscopy (STEM-EDS) revealed that atomic Pt was homogeneously dispersed over the whole material (Fig. 2f).

We then investigated the chemical configuration and binding status of Pt-SAs/MoS$_2$ by XPS (Fig. 2g and Supplementary Fig. 6). Two peaks at 72.0 and 75.3 eV were recorded in the Pt 4f XPS spectrum of Pt-SAs/MoS$_2$, indicative of Pt atoms with a partially

positive charge due to the mutual interactions between the isolated Pt atoms and the ce-MoS$_2$ support. The electronic and coordination structures of Pt species in Pt-SAs/MoS$_2$ were further confirmed by X-ray absorption spectroscopy, along with the Pt foil and PtO$_2$ as a comparison. The Pt L$_3$-edge X-ray absorption near-edge structure (XANES) analysis showed that the white line intensity for Pt-SAs/MoS$_2$ was obviously higher than that for the Pt foil (Fig. 2h), indicating that the Pt species in Pt-SAs/MoS$_2$ were partially oxidized. The positive oxidation state (1.7, inset of Fig. 2h) could be ascribed to the formation of Pt—S bond, resulting in the electron transfer from Pt to S. The extended X-ray absorption fine structure (EXAFS) spectra showed that no appreciable Pt—Pt bond (2.76 Å) was detected in Pt-SAs/MoS$_2$ (Fig. 2i), confirming that no Pt nanoparticles or clusters existed. A dominant peak in Pt-SAs/MoS$_2$ was observed at 2.26 Å, which corresponds to the Pt–S bond with a coordination number of *ca.* 3.2 (Supplementary Fig. 6, and Supplementary Table 5)[22]. The theoretical Pt–S distance (2.218, 2.218, and 2.467 Å) derived from the optimized Pt–Mo atop model fits well with our EXAFS data (mean bond length 2.26 Å). No first shell of Pt–Mo interaction excludes the possibility of covalent attachment of Pt atoms on S vacancies or Mo edge sites, in agreement with the magnified HAADF-STEM image (Fig. 2c). The strong bonding interaction between Pt atoms and neighboring S atoms in the support not only protects the single atoms from aggregation but also helps to regulate the electronic structure of single-atom metals.

**Generality of the SSED technique**. To demonstrate the versatility of the SSED method, we used various supporting TMD substrates, as well as precursor metals of interest, to construct a library of single-atom catalysts (Fig. 3a). We observed that Cu adatoms can be underpotentially deposited on different TMD substrates (ce-WS$_2$, ce-MoSe$_2$, ce-WSe$_2$, and ce-MoTe$_2$; Fig. 3b). The HAADF-STEM images of Pt-SAs/WS$_2$, Pt-SAs/MoSe$_2$, and Pt-SAs/WSe$_2$ confirm the predominance of single Pt atoms on the supports (Fig. 3c and Supplementary Fig. 7). We also extended the SSED method to the fabrication of other single-atom metals (Pd, Rh, Pb, Sn, and Bi; Fig. 3b and Supplementary Figs. 8–10), implying the broad applicability of this methodology and great potential for various heterogenous catalytic applications.

Bridging the gap between laboratory studies and industrial applications of materials remains challenging. To fulfill the requirements for large-scale industrial settings, we attempted to design a model experimental apparatus for producing Pt-SAs/MoS$_2$ in excellent yield (Supplementary Fig. 11). In addition, the loading of the single-atom metal can be precisely adjusted by controlling the time of Cu UPD or the number of active sites in the support (Supplementary Fig. 12); the precise and controlled growth of single-atom catalysts can be achieved within minutes.

**Mechanism of single-atom immobilization**. To better understand the process of site-specific UPD on the TMD materials, we tried to correlate the underpotential shift ($\Delta U_p$) with physical parameters pertinent to the system, which might semiquantitatively explain the energy gain for the energetically favored UPD compared to bulk deposition (Supplementary Note 4). We note that $\Delta U_p$ is linearly proportional to the absolute value of single-atom–support bonding ($\Delta G_{BE}$ in Supplementary Tables 6 and 7; Supplementary Figs. 13–15). We reason that the binding interaction arising from the specific affinity of metal adatoms to the substrate atoms is responsible for the energetically favored single-atom deposition of a metal on TMD materials (Fig. 3e). An explanation of the UPD on TMDs is therefore provided, which emphasizes the fact that this phenomenon is also general, where

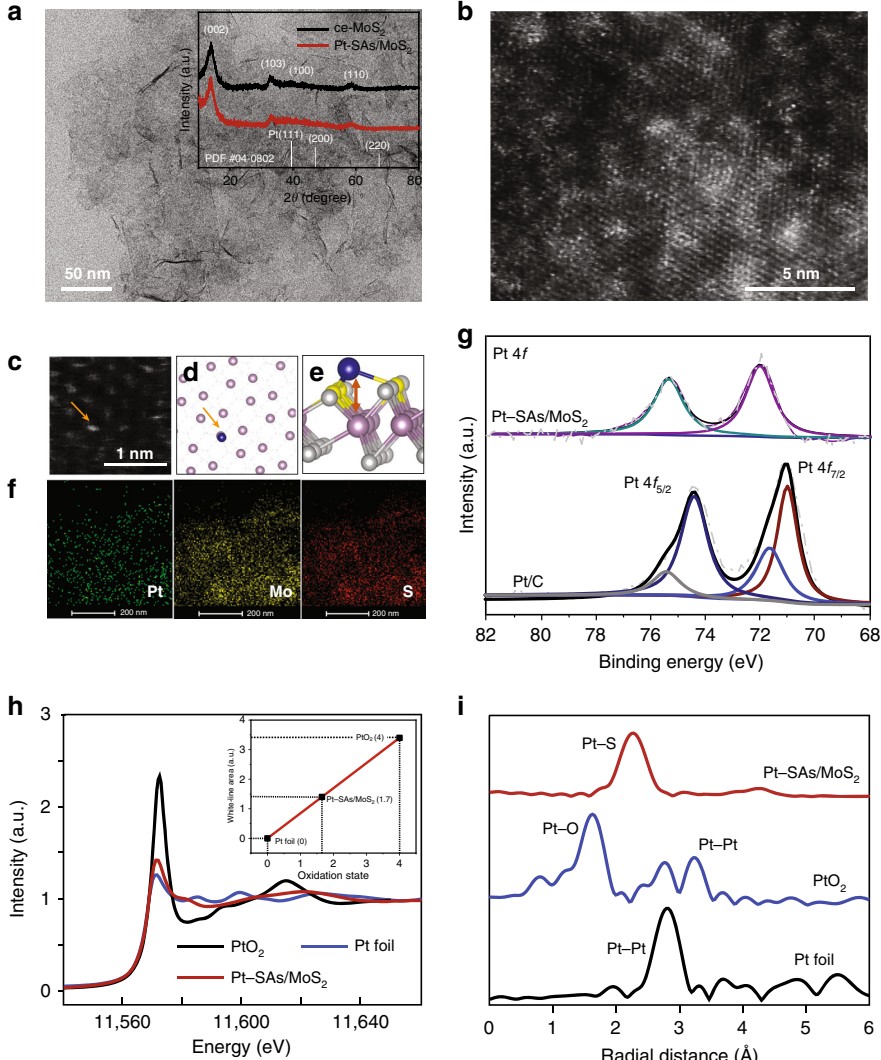

**Fig. 2 Structural characterization of Pt-SAs/MoS₂. a** Conventional TEM images of a freshly prepared Pt-SAs/MoS₂ sample. The inset shows the XRD patterns of ce-MoS₂ and Pt-SAs/MoS₂ samples. **b** HAADF-STEM image of Pt-SAs/MoS₂ (single Pt atoms highlighted by red circles). **c** Magnified HAADF-STEM image of Pt-SAs/MoS₂ and its corresponding top (**d**) and front (**e**) views of DFT-optimized structural model (purple: Mo; gray: S; yellow: S for the Pt attachment; blue: Pt), showing a typical environment of a Pt atom at sites beyond the Mo top site. The focal position of gray dash line in Fig. 2d represents the S atom. **f** Mapping images of Pt, Mo, and S atoms. **g** Pt 4 f XPS spectra of Pt-SAs/MoS₂ and commercial Pt/C. **h** The normalized XANES spectra of Pt foil, PtO₂, and Pt-SAs/MoS₂ at the Pt L₃-edge. The inset shows the average oxidation state of Pt in Pt-SAs/MoS₂. **i** Corresponding EXAFS profiles from Fig. 2h.

the binding energy of the adsorbed atoms on the substrate exceeds that in the respective bulk crystal.

In the self-terminating process of Cu UPD on the ce-MoS₂ nanosheets, the single-atom deposition reached saturation rapidly and was not dependent on the concentration of metal precursor (Supplementary Fig. 16). Moreover, the excessively high concentration of metal precursor did not lead to the formation of single-atom aggregation or nanoparticles (Supplementary Table 8). These results highlight the intelligence of SSED for single-atom synthesis, which is capable of finding the termination point of the deposition reaction.

To further interpret the changes in electron state and electron–phonon interactions during the process of Cu UPD on ce-MoS₂[23,24], we used operando Raman spectroscopy and electrochemical techniques to monitor the in situ growth of single-atom Cu (Supplementary Fig. 17). The notable change in the intensity and position of J₃ peak suggests perturbation of the 1 T vibration modes, particularly the vibration from the S atoms induced by the covalent attachment of Cu adatoms[25,26]. We then conducted an additional control experiment to confirm that

immobilization is not mediated by the weak noncovalent interactions of physisorption[17,25] (Supplementary Fig. 18). We also proposed using Lewis acid-base chemistry to elucidate the plausible coordinative interaction between Cu²⁺ ions and S atoms prior to single-atom immobilization[27] (Supplementary Note 5 and Supplementary Fig. 19).

All of these experiments demonstrate the critical role of the applied potential and S atoms for effective single-atom dispersion and stabilization. During the process of site-specific UPD, these two important factors provide energetically favored metal–support bonds and activation energy, respectively, for the self-terminating growth of atomically dispersed metal in terms of both thermodynamics and kinetics (Fig. 3e).

**Electrochemical HER study**. Encouraged by the successful fabrication of Pt-SAs/MoS₂, we selected the hydrogen evolution reaction (HER) as a model reaction for evaluating the electrocatalytic activity of single Pt atoms. Compared to ce-MoS₂, Pt-SAs/MoS₂ showed exceptional electrocatalytic activity in the HER

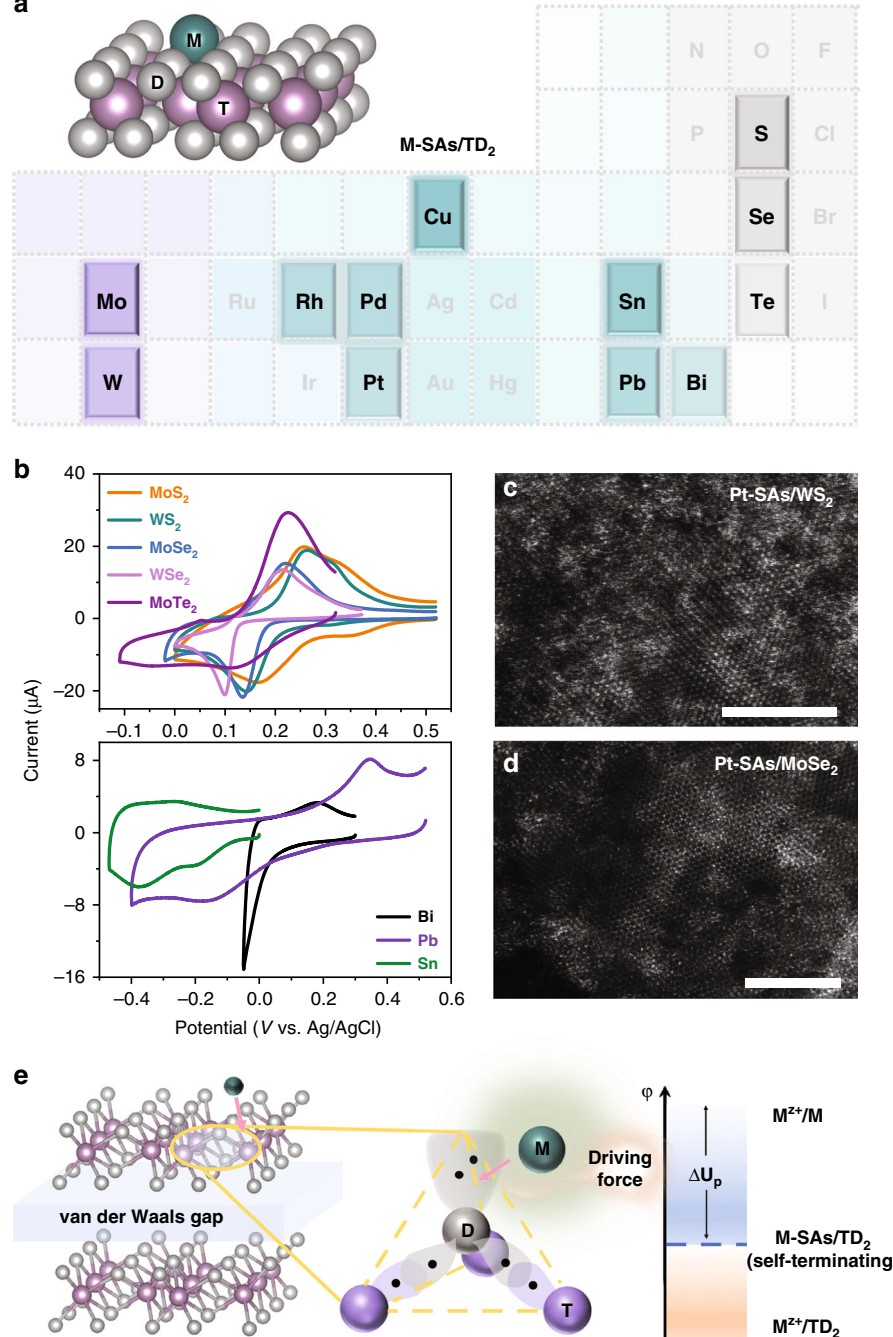

**Fig. 3 Generality and mechanism of SSED. a** A library of ADMCs synthesized on TMDs. **b** Cyclic voltammograms of UPD of Cu on different TMDs (top) and of UPD of different metals on ce-MoS$_2$ nanosheets (bottom). **c, d** Magnified HAADF-STEM image of Pt-SAs/WS$_2$ and Pt-SAs/MoSe$_2$ (scale bars: 5 nm). **e** Mechanism of SSED growth for ADMCs on TMDs. M represents the deposited single-atom metal; TD$_2$ represents the TMD substrate.

with negligible overpotential (Fig. 4a). Specifically, the overpotential required to achieve a current density of 10 mA cm$^{-2}$ for Pt-SAs/MoS$_2$ was only ~59 mV, which remarkably surpasses most of the state-of-the-art single-atom electrocatalysts (Supplementary Table 9). Pt-SAs/MoS$_2$ shows strikingly higher electrocatalytic HER activity than commercial Pt/C (Fig. 4a), a bulk Pt electrode, and aggregated Pt nanoparticles (Pt-NPs/MoS$_2$; Supplementary Fig. 20). For instance, as normalized to the loading amount of Pt, the mass activity of Pt-SAs/MoS$_2$ at an overpotential of 0.05 V was 17.14 A mg$^{-1}$, exceeding that of commercial Pt/C by a factor of 114 (inset of Fig. 4a). The catalytic production of H$_2$ by Pt-SAs/MoS$_2$ was further verified by gas

chromatography analysis (Supplementary Fig. 21). Electrochemical impedance spectroscopy (EIS) further explains the superior HER kinetics for Pt-SAs/MoS$_2$ compared to pure ce-MoS$_2$, which implies a much faster Faradaic process originating from the introduction of single-atom Pt (Supplementary Fig. 21).

To investigate the mechanistic insights into the outstanding electrocatalytic activity of Pt-SAs/MoS$_2$ in the HER, we evaluated the catalysis kinetics from Tafel plots (Fig. 4b)[28]. Pt-SAs/MoS$_2$ exhibits a Tafel slope of 31 mV dec$^{-1}$, whereas ce-MoS$_2$ exhibits a much larger Tafel slope of 91 mV dec$^{-1}$, implying faster kinetics owing to single-atom Pt decoration. The Tafel slope demonstrates that the HER on Pt-SAs/MoS$_2$ undergoes a Volmer–Tafel

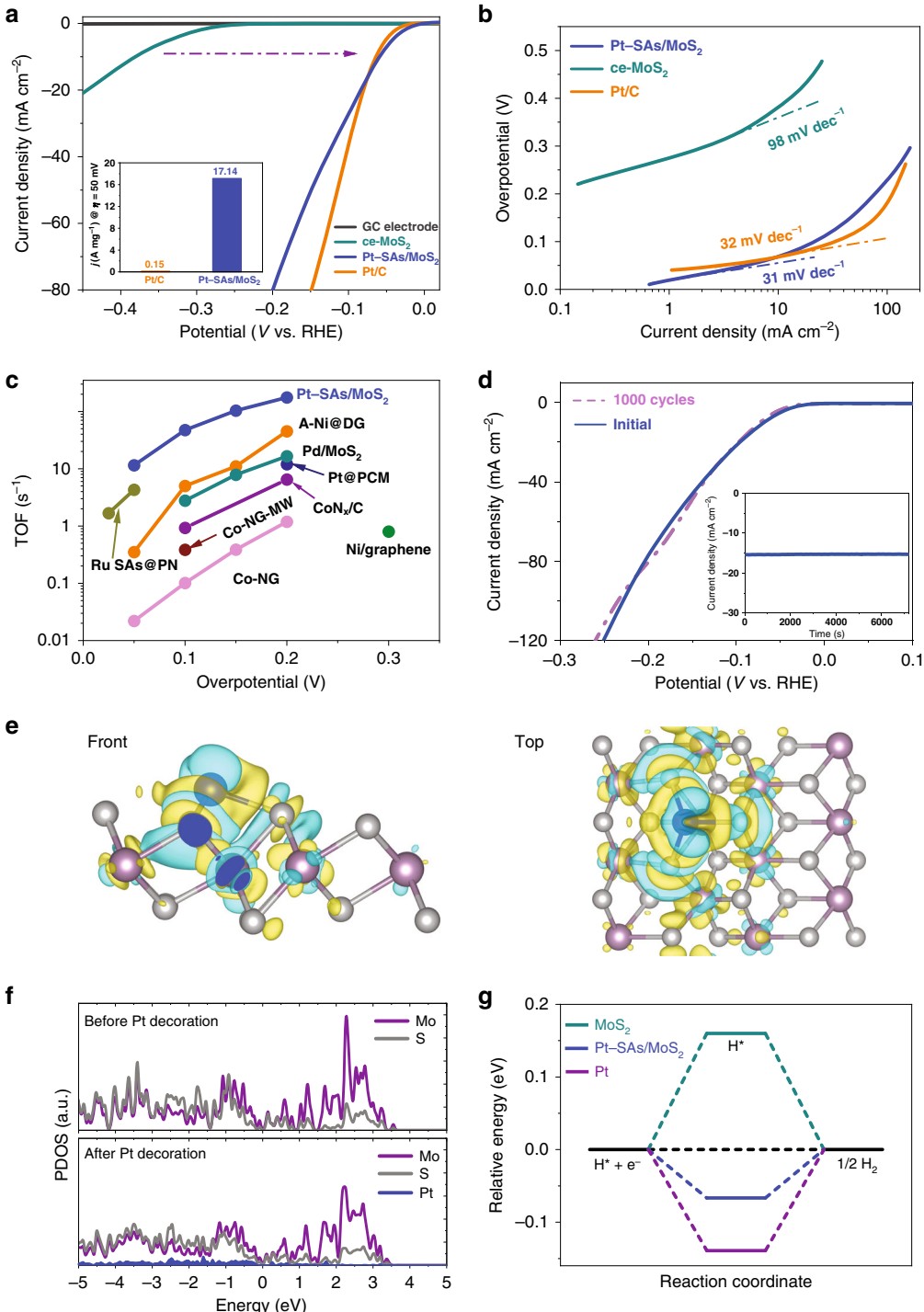

**Fig. 4 Electrocatalytic performance of the Pt-SAs/MoS$_2$ catalyst in the HER. a** HER polarization curves of bare, ce-MoS$_2$-, commercial Pt/C-, and Pt-SAs/MoS$_2$-covered GC electrodes in 0.5 M H$_2$SO$_4$ solution. Inset: the mass activity of Pt-SAs/MoS$_2$ normalized to the Pt loading at an overpotential of 50 mV in comparison with commercial Pt/C. **b** Tafel plots derived from the corresponding polarization curves. **c** The turnover frequency (TOF) curve of Pt-SAs/MoS$_2$ and a comparison with previously reported data for single-atom HER catalysts. **d** Stability test of Pt-SAs/MoS$_2$ by potential cycling before and after 1000 cycles. Inset: time-dependent current density curve of Pt-SAs/MoS$_2$ obtained at a constant overpotential of $\eta = 80$ mV for 2 h in 0.5 M H$_2$SO$_4$ solution. **e** Front (left) and top (right) views of the differential charge density of Pt-SAs/MoS$_2$. The yellow and blue surfaces correspond to gain and loss of charge, respectively. **f** Calculated PDOS of MoS$_2$ and Pt-SAs/MoS$_2$ with aligned Fermi levels at 0 eV. **g** Calculated the free energy diagram of the HER at the equilibrium potential for MoS$_2$, Pt-SAs/MoS$_2$, and Pt (pH 0).

pathway and that the Tafel recombination process is the dominant rate-limiting step (Supplementary Note 6, and Supplementary Fig. 22)[29]. The Pt-SAs/MoS$_2$ catalyst delivers an extremely high exchange current density of 2.24 mA cm$^{-2}$, which is largely increased by a factor of 1.6 compared to commercial

Pt/C (Supplementary Fig. 21). With the introduction of thiocyanate ion (SCN$^-$), the HER activity on Pt-SAs/MoS$_2$ was dramatically decreased due to blocking of the active Pt sites (Supplementary Fig. 21), suggesting that the activity indeed stems from the Pt single atoms. The turnover frequency (TOF) value of

these active Pt sites on Pt-SAs/MoS$_2$ is 47.3 s$^{-1}$ (100 mV), which is considerably higher than those of reported single-atom electrocatalysts (Fig. 4c, Supplementary Note 7, and Supplementary Table 9).

The electrocatalytic durability of Pt-SAs/MoS$_2$ was evaluated by an accelerated degradation test. Performance similar to that of the initial measurement for Pt-SAs/MoS$_2$ after 1000 cycles or 2 h demonstrates the high long-term stability of Pt-SAs/MoS$_2$ (Fig. 4d), much better than that of commercial Pt/C (Supplementary Fig. 23). The recycled Pt-SAs/MoS$_2$ that underwent the accelerated degradation test bore no signs of aggregated Pt atoms or Pt-containing crystalline phases, demonstrating the high stability of the single-atom Pt attached to the ce-MoS$_2$ nanosheets (Supplementary Fig. 23).

Similarly, compared to state-of-the-art single-atom electrocatalysts, exceptional electrocatalytic activities in the HER were also measured for Pt-SAs/WS$_2$, Pt-SAs/MoSe$_2$, and Pd-SAs/MoS$_2$ (Supplementary Fig. 24). It should also be noted that this methodology for the deposition of single-atom metal on TMDs is highly repeatable, and six independent batches of material displayed consistent HER performance. The results further indicated the synthetic stability and repeatability of the SSED method.

We performed first-principles calculations to gain deeper atom-level insight into the electronic interactions between Pt and MoS$_2$ and the catalytic contribution of single-atom Pt immobilization on Pt-SAs/MoS$_2$ for HER. The difference in charge density (Fig. 4e, Supplementary Fig. 25) reveals a redistribution of the electronic structure. Approximately 0.09 electrons are transferred from the Pt atom to MoS$_2$ through the formation of a Pt–S bond based on Bader charge calculation. As illustrated by the projected density of states (PDOS, Fig. 4f), the hybridization between Pt (5$d$ orbitals) and the neighboring S atoms effectively improves the dominance of $d$-electron orbitals near the Fermi level (Supplementary Fig. 26), leading to enhanced catalytic activity. Pt-SAs/MoS$_2$, with a higher Fermi level and lower work function, shows higher electronic energy levels and enhanced capability to provide electrons (Supplementary Fig. 27), and hence enables better conductivity (also evidenced by the experimental EIS results). We also calculated the free energy ($\Delta G_H^*$) for hydrogen adsorption to further elucidate the thermodynamic improvement of HER after immobilization of single-atom Pt (Fig. 4g and Supplementary Fig. 28). Compared to that of pure MoS$_2$ (0.160 eV), the calculated $\Delta G_H^*$ for the Pt-SAs/MoS$_2$ is approximately −0.067 eV, representing more favorable HER behavior than the commercial Pt catalyst (−0.139 eV).

From the apparent Tafel slope value obtained in the electrochemical HER study, the electrocatalytic pathway at atomic Pt in Pt-SAs/MoS$_2$ resembles that of conventional metal nanocrystals. However, our XANES, XPS, and Bader charge analysis experimentally and theoretically show that the single Pt atoms are positively charged, leading to a much higher total unoccupied density of Pt 5$d$ states. We reason that the atomic-scale tailoring should unconventionally modulate the adsorption state of hydrogen atoms on the Pt atom of Pt-SAs/MoS$_2$ owing to the unoccupied Pt 5$d$ orbitals combined with H 1$s$ orbital (Supplementary Fig. 29)[30–32]. We conclude that the maximum number of adsorbed H atoms in Pt-SAs/MoS$_2$ is six per Pt atom. Overall, the strengths of hydrogen adsorption decrease with an increase in the number of adsorbed H atoms, resulting in a minimum value if six H atoms are adsorbed (Supplementary Note 8). Compared to Pt nanocrystal, the increased number of catalytic Pt sites for H adsorption, as well as the lower adsorption energy of the product H$_2$, decrease the effective barrier in the overall kinetics, which explains the fast HER kinetics and high TOF that were experimentally observed.

## Discussion

ED methods have been widely used for single-atom synthesis in recent years. However, these processes are usually either completed within hours (at least 10 h) or less controllable, which result in the formation of nanoclusters/nanoparticles at longer cycling time or higher metal precursor concentration[33–35] (detailed comparison shown in Supplementary Fig. 30, Supplementary Note 9, and Supplementary Table 10). In this work, we report a self-terminating ED method to enable the growth of single-atom metals. The applied potential is a critical parameter, which distinguishes ED from other deposition techniques. Controlling the potential restricted to UPD region—at which metal–support bonding predominates over metallic bonding—can lead to the adsorbed single-metal atoms on the isolated chalcogen atoms with inherent nucleophilicity. Once the surface-limited reaction (UPD) reaches saturation, the self-termination stops the continuous atom growth regardless of the metal precursor concentration or deposition time, which reinforces the only formation of single-atom metals. The electrochemical synthesis directly provides the electrons for reducing single-metal ions into their elemental states. The fast kinetics accelerates the single-atom dispersion process and hence gives a much higher efficiency compared to some time-consuming and procedurally demanding methods (Supplementary Table 3). In addition, many conventional synthetic methods have been used under high temperature and pressure, which results in safety concerns and much energy consumption on an industrial scale. On the contrary, our electrochemical system is energy-efficient and green because it operates under mild conditions.

TMDs materials (e.g., MoS$_2$, WS$_2$, MoSe$_2$) have recently emerged as promising and efficient electrocatalysts for HER[36–38]. With a properly designed system, these TMDs materials show excellent HER activities, largely narrowing the gap with commercial Pt[38]. A general consensus has been reached that the activity of pristine TMDs originates from their coordinatively unsaturated chalcogen atoms along the edges, while the basal planes are inert[37]. Here, we fully harnessed the chalcogen atoms to construct single-atom Pt, which holds great promise as an efficient strategy to activate and optimize the inert basal plane of pristine TMDs materials.

Although UPD phenomena on some other metal chalcogenides (e.g., CdS, CdSe, Pd$_3$P$_2$S$_8$) or thiol-reagent modified electrodes have been previously reported[18,19,39–41], they were used to estimate the number of catalytic active sites, probe the energy states of the semiconductors, or detect the metal ions of interest, respectively. Thus, apart from the supporting substrates (MoS$_2$, WS$_2$, MoSe$_2$, WSe$_2$) used in our work, we hypothesize that these reported substrates might be potentially used as the site-specific substrates for the synthesis of ADMCs. With the N, P, O, or halogen atoms also possessing lone pair electrons and high affinities for metals, some metal nitrides, metal phosphides, MXenes, layered double hydroxides might serve as the candidate substrates for site-specific UPD of single-atom metals. We reasoned that more functional ADMCs could be obtained by UPD of a single-atom nonnoble metal (for example, Ag, Cd, and Hg) onto the site-specific substrates and subsequent galvanic displacement with a more-noble single-atom metal (for example, Ir, Ru, and Au) (Fig. 3a). Considering that the formed Pt SA might be further utilized as the nucleation site for the secondary UPD of single-atom nonnoble metal, we also propose that the SSED technology might also be generally applied to the bottom-up precise synthesis of bi- and multimetallic single-metal-site or cluster catalysts for series of domino and tandem reactions.

## Methods

**Materials.** Molybdenum(IV) sulfide (MoS$_2$) powder, molybdenum(IV) selenide (MoSe$_2$) powder, tungsten(IV) sulfide (WS$_2$) powder, $n$-butyllithium in hexane

(2.5 M), rhodium(III) chloride ($RhCl_3$), potassium tetrachloropalladate(II) ($K_2PdCl_4$), and Nafion perfluorinated resin solution (5 wt% in a mixture of low aliphatic alcohols and water, containing 45% water) were purchased from Sigma–Aldrich (St. Louis, MO); aluminum oxide powder (0.05 CR) was purchased from BAIKOWSKI (USA); $NaBH_4$ and SnO were purchased from the First Reagent Factory (China); potassium tetrachloroplatinate(II) ($K_2PtCl_4$) and Pt/C (5 wt%) were purchased from Macklin (China); acetonitrile, polyvinylpyrrolidone (PVP), $CuSO_4$, $Pb(NO_3)_2$, and $Bi(NO_3)_3$ were purchased from Nanjing Chemical Reagent (China); iodine and tungsten(IV) selenide ($WSe_2$) powder were purchased from Aladdin Industrial Corporation (China). All aqueous solutions were prepared with Millipore water (18.2 MΩ cm resistivity).

**Synthesis of Pt-SAs/MoS₂.** Chemically exfoliated transition metal dichalcogenides (ce-TMDs) were synthesized by lithium intercalation method[16] (details shown in Supplementary Methods). For the synthesis of Pt-SAs/MoS₂, a glassy carbon electrode was first polished with aluminum oxide powder, followed by ultrasonic cleaning in water and drying at room temperature. A working electrode was made by drop-casting 5 μL of the ce-MoS₂ suspension to cover a glassy carbon electrode (3-mm diameter). Cu atoms were first underpotentially deposited on the ce-MoS₂ nanosheets by controlling the potential at +0.10 V vs Ag/AgCl, at which only the UPD process occurs, in an Ar-saturated solution of 0.1 M $H_2SO_4$ containing 2 mM $CuSO_4$. The product was termed Cu-SAs/MoS₂. After being washed with water, the product Cu-SAs/MoS₂ was immediately transferred into an Ar-saturated solution of 0.05 M $H_2SO_4$ containing 5 mM $K_2PtCl_4$. The electrode was left in this solution for more than 20 min at an open-circuit potential to ensure the complete galvanic replacement of Cu by Pt(II) to form the atomically dispersed, Pt-modified ce-MoS₂ nanohybrid (termed Pt-SAs/MoS₂). The resulting product was transferred into an argon-filled glovebox for storage at room temperature. All electrochemical measurements were performed on a CHI 660E Instrument (Chenhua, China) at room temperature. Other single-atom catalysts were prepared according to similar procedures as Pt-SAs/MoS₂, instead of using different ce-TMDs (ce-WS₂ and ce-MoSe₂) as the supporting substrates or metal precursors ($K_2PdCl_4$, $RhCl_3$, SnO, Bi $(NO_3)_3$, and $Pb(NO_3)_2$), respectively (details shown in Supplementary Methods).

**Physical characterization.** Zeta potentials of ce-MoS₂ and Pt-SAs/MoS₂ were measured by using a Malvern Instruments Zetasizer Nano-ZS90 (phosphate-buffered saline, pH 7.4). Inductively coupled plasma optical emission spectrometry (ICP-OES) was used with a CHN-O-Rapid elemental analyzer (Heraeus, Germany) to determine the concentration of single-atom metals. The samples for ICP-OES analysis were treated with aqua regia in Teflon-lined autoclaves at 230 °C for 12 h. The content of Pt in Pt-SAs/MoS₂ remained the same, ~5.1 wt%, before and after HER catalysis. TEM (JEOL JEM-2100, Japan) and field emission electron microscopy (JEOL JEM-2800, Japan) were used to characterize the morphologies and element maps of the catalysts. High-angle annular dark-field-scanning transmission electron microscopy (HAADF-STEM) was carried out on an FEI Titan[3] G2 60-300 system equipped with double aberration correctors and operated at 80 kV. XRD patterns (X'TRA, ARL, Switzerland) were collected to determine the crystal structures of the samples. XPS measurements were performed on a PHI 5000 VersaProbe (Japan) with the calibration of binding energies based on the C 1 s peak energy of 284.6 eV. The X-ray absorption spectroscopy at the Pt $L_3$-edge was obtained at the BL14W1 beam line of the Shanghai Synchrotron Radiation Facility (SSRF), using a Si(111) double-crystal monochromator operated at 3.5 GeV. Pt foil and $PtO_2$ were used as reference samples and measured in the transmission mode. Pt-SAs/MoS₂ was measured in fluorescence mode. All the spectra were analyzed using the Athena and Artimas software (0.9.25 version)[42]. Raman spectra were collected on a LabRAM Aramis Raman spectrometer (HORIBA, Ltd., Japan).

**Electrochemical measurements for HER.** Linear sweep voltammetry with a scan rate of 2 mV s$^{-1}$ was conducted using a CHI 660E instrument (Chenhua, China) in 0.5 M $H_2SO_4$ using a Ag/AgCl electrode (saturated KCl) as the reference electrode, a graphite rod as the counter electrode, and a glassy carbon electrode as the working electrode. The Ag/AgCl electrode was calibrated with respect to the reversible hydrogen electrode (RHE). In 0.5 M $H_2SO_4$, $E_{RHE} = E_{Ag/AgCl} + 0.2220$ V. Impedance measurements were performed at frequencies ranging from 0.1 to 100 kHz with an amplitude of 10 mV at the onset potential of each electrocatalyst, for which the current density was 0.5 mA cm$^{-2}$.

The reaction product hydrogen was measured using a gas chromatograph (GC-2014, Shimadzu) equipped with a separation column (MS-13X, 80/100 mesh, 3.2 × 2.1 mm × 2.0 m) and a thermal conductivity detector. Argon gas was used as the carrier gas in the chromatograph. The following parameters were used: column temperature, 80 °C; detector temperature, 100 °C; and bridge current, 60 mA.

The turnover frequency (TOF) of the catalyst was calculated according to the following equation:

$$TOF = I/(2F \times n),\tag{1}$$

where $I$ is the measured current during the linear sweep measurement, $F$ is the Faraday constant (96,500 C mol$^{-1}$) and $n$ is the molar amount of active Pt sites. The factor 1/2 represents that two electrons are required to form one molecule of

hydrogen ($2H^+ + 2e^- \rightarrow H_2$). Detailed calculations are shown in Supplementary Note 7.

**DFT calculations.** We performed first-principle calculations by using the Vienna ab initio simulation package (VASP 5.4.4) with the Perdew–Burke–Ernzerhof generalized gradient approximation for the exchange-correlation functional and the projector augmented wave method. The DFT-D2 method was adopted to describe the van der Waals interactions between the adsorbed atoms and the support. We used a $2 \times 3 \times 1$ supercell of $MX_2$ (X = S, Se; M = Mo, W) and a vacuum region of 18 Å to eliminate interactions between the neighboring cells of slab models for all calculations. The plane-wave cutoff energy was set as 400 eV for all calculations; $3 \times 3 \times 1$ and $5 \times 5 \times 1$ Γ-centered k points were adopted for geometry optimization and self-consistent electronic calculation, respectively. All the atoms were relaxed until the residual force was <0.01 eV/Å and the self-consistent field tolerance level was $1.0 \times 10^{-5}$ a.u. for the geometry optimizations.

For the geometry optimizations of Cu-SAs/MoS₂, we chose a $2 \times 3 \times 1$ supercell of MoS₂ and then placed one Cu atom on four inequivalent adsorption sites (Mo1, Mo2, S1, and S2). The formation energy of the Cu adsorption was calculated as follows:

$$E = E_{Cu-SAs/MoS_2} - E_{MoS_2} - E_{Cu}.\tag{2}$$

We found that the structure in which Cu is adsorbed on the Mo1 site was the most stable with the lowest energy of formation. Similar phenomena were observed for Cu-SAs/MoSe₂ and Cu-SAs/WS₂ with the Mo1 site being the most stable adsorption position. Subsequently, we substituted the Cu atoms with Pt and found that structures having a Pt atom bound at the Mo1 site were still the most stable for Pt-SAs/MoS₂, Pt-SAs/WS₂, and Pt-SAs/MoSe₂.

The differential charge density (Fig. 4e) was plotted according to the charge density distributions of MoS₂ and Pt-SAs/MoS₂ (Supplementary Fig. 25) as follows:

$$\Delta\rho = \rho_{Pt-SAs/MoS_2} - \rho_{MoS_2} - \rho_{Pt}.\tag{3}$$

The free energy of the adsorbed state was calculated as follows:

$$\Delta G_{H^*} = \Delta E_{H^*} + \Delta E_{ZPE} + T\Delta S,\tag{4}$$

where $\Delta E_{H^*}$ is the energy of hydrogen adsorption, and $\Delta E_{ZPE}$ is the difference in zero-point energy between the adsorbed states and the gas phase. The hydrogen adsorption site was confirmed by comparing the energy of hydrogen adsorption at different positions on the catalyst surface after free geometry optimization with an optimization tolerance level of $2.0 \times 10^{-5}$ a.u. (Supplementary Fig. 28). The van der Waals correction was taken into consideration in all these calculations.

## Data availability

All data are available from the authors, please refer to author contributions for specific data sets. Source data are provided as a Source Data file. Source data are provided with this paper.

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

## Acknowledgements
We thank the Shanghai Synchrotron Radiation Facility (14W1, SSRF) for assistance in structural characterizations. We thank Dr. Ting-Ting Zhai, Ms. Wei-Xuan Xu, Ms. Xue-Yan Shao, and Mr. Shuai Su for assistance in electrochemical data analysis. We thank Dr. Yu Wang for assistance in structural analysis. We thank Dr. Xiao-Kun Huang for his useful discussion on DFT simulations. This work was supported by grants from the National Key Research and Development Program of China (2017YFA0206500), the National Natural Science Foundation of China (21902076, 21635004), the Natural Science Foundation of the Jiangsu Province (BK20190289), and the Excellent Research Program of Nanjing University (ZYJH004).

## Author contributions
Prof. X.X. and Y.S. initiated the project and conceived the experiments. Prof. X.X., Y.S., and W.H. designed the experiment and collected data. Y.S., J.L., Y.Z., Z.L., and Y.Y. carried out all the electrochemical experiments and data analysis. Y.S. wrote the manuscript together with W.H. and Prof. X.X. All the authors contributed to and commented on this paper.

## Competing interests
The authors declare no competing interests.
