## [Peer Review File · Nature Communications]

Reviewers' Comments:

Reviewer #1:

Remarks to the Author:

In this work, the authors reported an electrodeposition method to synthesize various single atoms supported on different transition metal dichalcogenides (TMDs). A series of single atom catalysts have been successfully synthesized through this method, including Pt-SAs/MoS₂, Pt-SAs/MoSe₂, Pt-SAs/WS₂, Pd-SAs/MoS₂, Sn-SAs/MoS₂, Pb-SAs/MoS₂, and Bi-SAs/MoS₂. The resulted Pt-SAs/WS₂ also shows excellent performance for HER. The characterizations of these materials are fine and the manuscript is well organized. However some issues exist as follows.

1. The major issue is the difference between this work and the newly published work in Nature Commun. (Electrochemical deposition as a universal route for fabricating single-atom catalysts, Nature Commun. (2020) 11:1215 <https://doi.org/10.1038/s41467-020-14917-6>). In the newly published work, the TMDs supported single atom catalysts (for example, Ir₁/MoS₂) can also be synthesized by electrochemical deposition method.
2. Another major issue is the characterizations of ADMCs, like Pd-SAs/MoS₂, Sn-SAs/MoS₂, Pb-SAs/MoS₂, and Bi-SAs/MoS₂, some ultrasmall particles or clusters can be clearly observed in the TEM (Supplementary Figure 19b) or HAADF-STEM images (Supplementary Figure 21c, e, h), the author should re-check the metal atomic dispersion of these samples.
3. Minor issue: Is it capable of synthesizing single atoms on other substrates through this method? For example, metal oxide or CN.

Reviewer #2:

Remarks to the Author:

This manuscript present an interesting study of single atom catalyst for hydrogen evolution reactions (HER). The single atom catalyst has been an interesting topic for some time and a substantial number of papers have been published so far, including the electrochemical reactions. A good list of the works can be found in the Table 9 in the Supplementary of this manuscript. The strength of this work is that several metals has been considered and the synthesis method is quite simple. The characterization of the studied systems have been very careful and thus the Supplementary is 80 pages. Over all the material developed here has similar HER performance as the best current materials (Table 9 in Suppl.) The HER in acidic environment is not very important reaction but the work here is a good example of developing single metal catalyst for a challenging environments (solid-liquid interphase, very low pH) and it can be published in Nature Comm. after the text and some minor issues has been improved. The drawback of this is manuscript is the text. The manuscript is hard to read since it contain so many references to the Suppl. In addition, it is difficult to see what system have been studied. Fig 3 a) a map of the studied systems has been shown. There are some cyclic voltamograms shown (3 b) but it is unclear to which materials the HER has been studied. The Pd case is well presented and there are additional data in Suppl. but the main manuscript is vague. I see that the manuscript do not managed to describe well the research that is done. I suggest that the manuscript focus on the big picture and the Supplementary will have a starting chapter that collects the details together. Now these technical details are in the manuscript and this make it so hard to read.

In addition, the manuscript gives an impression that MoS₂ (and other metal dichalcogenides) are rather poor HER catalyst. They are not. In MoS₂ the catalytically active part are the flake edges and with properly designed systems the MoS₂ is almost as good HER catalyst as Pt. I am a bit surprised that this do not show in the experiments.

The fig 4g shows data form DFT calculations. To what site the H in MoS₂ has been attached. There are several low energy sites on the MoS₂ edges. The best (ΔG near 0) H binding sites should be reported and the site need to be reported.

Reviewer #3:

Remarks to the Author:

This is a very interesting and thorough study of the formation of atomically dispersed metal atoms by electrodeposition. I would ask that the reviewers consider the following points,

1. It would be interesting to know how well these catalysts compare with bulk Pt for the HER. This is probably more important than the comparison with MoS₂. For Fig. 4a, it would be helpful to show the iV curve for bulk Pt to allow comparison with the dispersed Pt. Also, for the Tafel plot in Fig. 4b and the plots in SI Fig. 33, in addition to the Tafel slopes, which tell something about the mechanism, it would be good to tell the exchange current density, which is a direct measure of the catalytic activity. It would be surprising if these catalysts were better than bulk Pt.
2. Is it reasonable to be able to see single Pt atoms by HAADF-STEM, as suggested in Fig. 2c?
3. For Fig. 2i, I believe that the x-axis should be "Radial distance" not "Radical distance"
4. Can you estimate the coverage of Cu in the UPD from the cyclic voltammograms? It is truly less than a monolayer?
5. The CVs are not as sharp as usually observed for UPD. Is there an explanation for this? Would it be possible to plot the CVs using current density instead of current?

Reviewer #1 (Remarks to the Author):

In this work, the authors reported an electrodeposition method to synthesize various single atoms supported on different transition metal dichalcogenides (TMDs). A series of single atom catalysts have been successfully synthesized through this method, including Pt-SAs/MoS₂, Pt-SAs/MoSe₂, Pt-SAs/WS₂, Pd-SAs/MoS₂, Sn-SAs/MoS₂, Pb-SAs/MoS₂, and Bi-SAs/MoS₂. The resulted Pt-SAs/WS₂ also shows excellent performance for HER. The characterizations of these materials are fine and the manuscript is well organized. However some issues exist as follows.

Response: We thank the reviewer for his/her positive comments. The suggestions are very valuable and constructive to helping us to enhance the quality of our manuscript. We have very carefully revised the manuscript and replies to the comments point-by-point.

1. The major issue is the difference between this work and the newly published work in Nature Commun. (Electrochemical deposition as a universal route for fabricating single-atom catalysts, Nature Commun. (2020) 11:1215 <https://doi.org/10.1038/s41467-020-14917-6>). In the newly published work, the TMDs supported single atom catalysts (for example, Ir1/MoS₂) can also be synthesized by electrochemical deposition method.

Response: We thank this reviewer for this valuable comment and concern. As the reviewer mentioned, Zeng's group have reported an universal electrodeposition method for the fabrication of single-atom metals [*Nat. Commun.* 11, 1215 (2020)]. Very interestingly, the electrodepositions can be both cathodically and anodically conducted for synthesis of single-atom metals with distinct electronic states, which holds great promises for various catalytic reactions. **The major difference of our work and their approach is technique basis: they used the bulk electrodeposition, while we carried out the underpotential deposition (UPD).** During UPD process, **owing to the applied potential more positive than equilibrium potential of metal ions, only metal-support interaction is energetically favorable, whereas metal-metal bonding is forbidden.** In our work, underpotential deposition on site-specific substrate enables the automatically terminating growth of single-atom metals, **which is not**

limited by the deposition time or the metal precursor concentration. This intrinsically self-terminating effect distinguishes our site-specific UPD method from other previously reported electrodeposition method that is less controllable in deposition of single atoms, providing a potentially “**smarter**” **methodology** for single-atom synthesis.

In order to better elucidate the difference between our work and previously reported ones, we have provided the detailed comparisons in the revised Supplementary Information (Supplementary Note 9, Supplementary Figure 30, and Supplementary Table 10). We have also added this valuable reference and revised the corresponding description to avoid any confusion (see Discussion paragraph 1 in the revised manuscript). The revised details are also shown below.

Electrodeposition offers a facile, controllable and room-temperature method for reducing metal ions into their elemental states. In this case, an efficient electrodeposition method has been widely used for single-atom synthesis in recent years [*ACS Catal.* **7**, 3121 (2017); *Angew. Chem. Int. Ed.* **56**, 13694 (2017); *Chem. Mater.* 2019, 31, 2, 429; *Nat. Commun.* **10**, 1743 (2019);]. The cathodic deposition of single-atom metals on a working electrode can be achieved by anodic dissolution of a bulk metal foil electrode as a counter electrode under acidic conditions using a three-electrode configuration (Fig. R1). However, the potential cycling process is usually completed within hours (at least 10 h) and is less controllable, which results in the formation of nanoclusters or nanoparticles at longer cycling times.

Fig. R1 Traditional single-atom electrodeposition mechanism.

The paper from Zeng’s group reported a universal and rapid electrodeposition approach for the fabrication of single-atom metals [*Nat. Commun.* **11**, 1215 (2020)]. They proposed that the electrodeposition process resembles the molecular nucleation mechanism. The upper limit of mass loading for single-atom metals cannot exceed the minimum supersaturation level, otherwise single-atom metal tends to nucleate (Fig. R2). Thus, single-atom synthesis can be realized by controlling the metal precursor concentration and deposition time.

Fig. R2 Cathodic/anodic electrodeposition (C, A-ED) mechanism of single-atom metals reported by Zeng’s group (taking cathodic deposition as an example).

In our work, we developed an “intelligent” methodology for the single-atom growth by site-specific electrodeposition which is capable of automatically terminating the aggregation of metal atoms (Fig. R3). Such intrinsically self-terminating effect distinguishes our site-specific UPD method from other previously reported electrodeposition method for single-atom synthesis. We show that the site-specific UPD method can be used to produce high-loading single-atom metals without the consideration of the high metal precursors or long deposition time (Supplementary Table 10), which might seriously cause atom aggregation in other electrodeposition methods. After the formation of thermodynamically favorable metal–support bonds, the sequential formation of metal–metal bond is forbidden at the UPD potential, restricting it to the single-atom metal. In our design, two requisite factors should be considered: (1) identifying electrically conductive substrate materials, which consist of isolated active sites that possess lone pair electrons and suitable electronegativity for the UPD of single atoms; (2) choosing suitable applied electrodeposition potential restricted to UPD region, at which metal–support bonding predominates over metallic bonding. Our site-specific UPD method for single-atom synthesis can be rapidly completed on a timescale of seconds to minutes. Additionally, we confirm that single-atom metals are straddled atop Mo by coordinating with three nearest neighboring S of Mo (see HADDF-STEM image, EXAFS data, and DFT simulations), which shows distinct from depositing site of vacancies/edges/defects reported by previous works.

Fig. R3 Self-terminating growth mechanism of single-atom metals reported by our group.

We further list a Supplementary Table 10 to compare the electrodeposition methods for single-atom synthesis:

Comparison \ Methods	Traditional single-atom electrodeposition	C, A-ED method reported by Zeng's group	Our SSED method
Loading amount	0.2~1.6 wt%	2.3 wt%	5.1 wt%
Time scale	>10 hours	minutes	minutes
Controlling deposition time	Need	Need	No need
Controlling metal precursor concentration	No need	Need	No need
Electrolyte	H ₂ SO ₄	H ₂ SO ₄ /KOH containing metal precursor	H ₂ SO ₄ containing metal precursor
Depositing site	Vacancies/edges/steps (into lattice)	Vacancies/edges/steps (into lattice)	Atop Mo (coordinating with three nearest neighbouring S)
Capability of adjusting electronic states of metals	No	Yes	No

2. Another major issue is the characterizations of ADMCs, like Pd-SAs/MoS₂, Sn-SAs/MoS₂, Pb-SAs/MoS₂, and Bi-SAs/MoS₂, some ultras-mall particles or clusters can be clearly observed in the TEM (Supplementary Figure 19b) or HAADF-STEM images (Supplementary Figure 21c, e, h), the author should re-check the metal atomic dispersion of these samples.

Response: We really thank the reviewer for pointing this out. We are sorry about the imperfect characterization of some ADMCs in the original text. We have very carefully re-characterized those ADMCs with freshly prepared samples (Pd-SAs/MoS₂, Rh-SAs/MoS₂, Sn-SAs/MoS₂, Pb-SAs/MoS₂, and Bi-SAs/MoS₂) to confirm the atomic dispersion of these samples (see updated data in Supplementary Figs. 8 and 10).

Supplementary Figure 8. Characterization of the morphology of Pd-SAs/MoS₂ and Rh-SAs/MoS₂. (a) A conventional TEM image of a freshly prepared Pd-SAs/MoS₂ sample. Inset: the corresponding SAED pattern. No obvious clusters or nanoparticles are found in Pd-SAs/MoS₂, implying that most of the Pd exists in the atomically dispersed form. (b) HAADF-STEM image of Pd-SAs/MoS₂ (scale bar: 5 nm). (c) EDX

mapping images of Pd, Mo, and S elements. (d) HAADF-STEM image of Rh-SAs/MoS₂ (scale bar: 5 nm). Inset shows the conventional TEM images of a freshly prepared Rh-SAs/MoS₂ sample. (e) Rh 3d XPS spectrum of Rh-SAs/MoS₂. The core-level binding energy of Rh 3d 5/2 in Rh-SAs/MoS₂ is located at *ca.* 307.6 eV. Considering that the XPS Rh⁰ peak is located at *ca.* 307.2 eV, the Rh⁺ peak at *ca.* 307.5 eV, and the Rh³⁺ peak at *ca.* 308.3 eV, the Rh atoms in Rh-SAs/MoS₂ are positively charged and have an oxidation state of *ca.* +1.

Supplementary Figure 10. (a, c, e) Conventional TEM images of freshly prepared Sn-SAs/MoS₂, Pb-SAs/MoS₂, and Bi-SAs/MoS₂ samples. (b, d, f) Magnified HAADF-STEM images of Sn-SAs/MoS₂, Pb-SAs/MoS₂, and Bi-SAs/MoS₂ samples (scale bar: 2 nm).

3. Minor issue: Is it capable of synthesizing single atoms on other substrates through this method?
For example, metal oxide or CN.

Response: We thank the reviewer for the question and interest. We have also successfully synthesized single-atom metals on some other substrates through this method, for example, some carbon-based materials (*e.g.* hetero-atom doped graphene, unpublished data shown in Fig. R4). Furthermore, in view of the growth mechanism (atoms possessing lone pair electrons and high affinities for metals), some more potential substrates have been highlighted in the discussion section (for example, metal nitrides, metal phosphides, MXenes, layered double hydroxides), which are undergoing in our group.

Fig. R4 HADF-STEM image of single-atom metal on the hetero-atom doped graphene substrate.

Reviewer #2 (Remarks to the Author):

This manuscript present an interesting study of single atom catalyst for hydrogen evolution reactions (HER). The single atom catalyst has been an interesting topic for some time and a substantial number of papers have been published so far, including the electrochemical reactions. A good list of the works can be found in the Table 9 in the Supplementary of this manuscript. The strength of this work is that several metals has been considered and the synthesis method is quite simple. The characterization of the studied systems have been very careful and thus the Supplementary is 80 pages. Over all the material developed here has similar HER performance as the best current materials (Table 9 in Suppl.) The HER in acidic environment is not very important reaction but the work here is a good example of developing single metal catalyst for a challenging environments (solid-liquid interphase, very low pH) and it can be published in Nature Communications after the text and some minor issues has been improved.

Response: We thank this reviewer for his/her positive comments. The suggestions are very valuable and constructive to helping us to enhance the quality of our manuscript. We have very carefully revised the manuscript and replies to the comments point-by-point.

1. The drawback of this manuscript is the text. The manuscript is hard to read since it contain so many references to the Suppl. In addition, it is difficult to see what system have been studied. Fig 3 a) a map of the studied systems has been shown. There are some cyclic voltamograms shown (3 b) but it is unclear to which materials the HER has been studied. The Pd case is well presented and there are additional data in Suppl. but the main manuscript is vague. I see that the manuscript do not managed to describe well the research that is done. I suggest that the manuscript focus on the big picture and the Supplementary will have a starting chapter that collects the details together. Now these technical details are in the manuscript and this make it so hard to read.

Response: We really thank the reviewer for this constructive advice. In this work, we focus on the big picture of **site-specific electrodeposition methodology (SSED) for the self-terminating growth of single-atom metals**. We used Fig. 3 to show the universality of our SSED for various single-atom syntheses. In order to better clarify

this big picture, we did not go through the HER performance and mechanism of all the materials as shown in Fig. 3; instead, we chose Pt-SAs/MoS₂ as a model system in the manuscript (Fig. 4) to present the potential application of the single-atom metals in heterogenous catalysis, and move the HER results of some other single-atom materials (such as single-atom Pd or single-atom Pt on other TMD supports) into the Supplementary .

To make the paper easier to read, we revised the paper from the following aspects:

1) following the reviewer's suggestion, we have carefully polished and re-merged many figures (see Supplementary Figs. 3-10, Supplementary Fig. 12, Supplementary Fig. 15, Supplementary Figs. 20 and 21, Supplementary Figs. 23 and 24, Supplementary Fig. 28, Supplementary Fig. 30) throughout the revised Supplementary Materials to reduce the references to Supplementary;

2) we have listed Supplementary Notes that collect all the technical details together. We have also moved many detailed description about the characterizations of single-atom metals and mechanism of single-atom immobilization into the Supplementary Notes (Supplementary Notes 4 and 5) or Supplementary Figure Legends (Supplementary Figs. 17 and 18), and further made a starting summary of the whole Supplementary Materials (Page 2 in Suppl.).

2. In addition, the manuscript gives an impression that MoS₂ (and other metal dichalcogenides) are rather poor HER catalyst. They are not. In MoS₂ the catalytically active part are the flake edges and with properly designed systems the MoS₂ is almost as good HER catalyst as Pt. I am a bit surprised that this do not show in the experiments.

Response: We thank the reviewer for this valuable suggestion, which makes our study become integrity and preciseness. We strongly agree with the reviewer that MoS₂ (and other metal dichalcogenides) themselves are good HER catalysts. We have added the

following part to the Discussion Section to avoid any confusion about the transition metal dichalcogenides.

“TMDs materials (*e.g.* MoS₂, WS₂, MoSe₂) have recently emerged as promising and efficient electrocatalysts for HER [*J. Am. Chem. Soc.*, 2005, 127, 5308; *Science*, 2007, 317, 100; *Chem. Soc. Rev.*, 2015, 44, 5148]. With properly designed systems, these TMDs materials show excellent HER activities, largely narrowing the gap with commercial Pt/C [*Chem. Soc. Rev.*, 2015, 44, 5148]. A general consensus has been reached that the activity of pristine TMDs originates from their coordinatively unsaturated chalcogen atoms along the edges, while the basal planes are inert [*Science*, 2007, 317, 100]. In our work, we fully harnessed the chalcogen atoms to construct single-atom Pt by using an electrodeposition method, which holds great promise as an efficient strategy to activate and optimize the inert basal plane of pristine TMDs materials.”

3. The fig 4g shows data from DFT calculations. To what site the H in MoS₂ has been attached. There are several low energy sites on the MoS₂ edges. The best (Delta G near 0) H binding sites should be reported and the site need to be reported.

Response: We really thank the reviewer for pointing this out. We have carefully re-checked and optimized the structure of hydrogen-adsorbed 1T-MoS₂ [refer to *ACS Catal.* 2016, 6, 4953; *Nat. commun.* 2018, 9, 2120], and revised its free energy in the manuscript. We also provide the DFT-calculated structures of H binding sites on MoS₂ and Pt-SAs/MoS₂ in Supplementary Fig. 28.

According to the revised Fig 4g, the optimized H binding site should be the basal plane of 1T-MoS₂. Unlike 2H-MoS₂ where the catalytic activity arises from the edges, experimental results from Chhowalla *et al.* suggested that the active sites of chemically exfoliated 1T MoS₂ nanosheets are mainly located on the basal plane and the contribution of the metallic edges to the overall HER efficiency is relatively small [*Nano Lett.* 2013, 13, 6222]. The much greater active surface area of 1T nanosheets

with respect to the edge portion thus guarantees the HER activity. Our optimized DFT-calculated results also agree well with the previously reported ones [refer to *ACS Catal.* 2016, 6, 4953; *Nat. commun.* 2018, 9, 2120].

We calculated the free energy (ΔG_{H}^*) for hydrogen adsorption to further elucidate the thermodynamic improvement of HER after immobilization of single-atom Pt (Fig. 4g and Supplementary Fig. 28). Compared to that of pure MoS₂ (0.160 eV), the calculated ΔG_{H}^* for the Pt-SAs/MoS₂ is approximately -0.067 eV, representing more favorable HER behavior than the commercial Pt catalyst (-0.139 eV).

Fig. 4g Calculated free energy diagram of HER at the equilibrium potential for MoS₂, Pt-SAs/MoS₂, and Pt (pH 0).

Supplementary Figure 28. (a) Top, front and perspective views of the DFT-calculated geometries for MoS₂ with an H atom adsorbed on the S top site. (b) Top, front and perspective views of the DFT-calculated geometries for Pt-SAs/MoS₂ with an H atom

absorbed on the Pt top site. The adsorption site of H is confirmed by comparing the energy of hydrogen adsorption at different positions on the catalyst surface after free geometry optimizations with an optimization tolerance level of 2.0×10^{-5} a.u. The atom color code is the same as in Supplementary Fig. 5.

Reviewer #3 (Remarks to the Author):

This is a very interesting and thorough study of the formation of atomically dispersed metal atoms by electrodeposition. I would ask that the reviewers consider the following points.

Response: We thank the reviewer for his/her positive comments. The suggestions are very valuable and constructive to helping us to enhance the quality of our manuscript. We have very carefully revised the manuscript and replies to the comments point-by-point.

1. It would be interesting to know how well these catalysts compare with bulk Pt for the HER. This is probably more important than the comparison with MoS₂. For Fig. 4a, it would be helpful to show the iV curve for bulk Pt to allow comparison with the dispersed Pt. Also, for the Tafel plot in Fig. 4b and the plots in SI Fig. 33, in addition to the Tafel slopes, which tell something about the mechanism, it would be good to tell the exchange current density, which is a direct measure of the catalytic activity. It would be surprising if these catalysts were better than bulk Pt.

Response: We thank the reviewer for these comments. Following the reviewer's advices, we have made the following revisions and comments:

- (1) We added the LSV curve, mass activity, and Tafel plot for bulk Pt (commercial Pt/C catalyst) to Fig. 4a and 4b for comparison with the dispersed Pt;
- (2) Exchange current density, which is a direct measure of the catalytic activity, is also shown in the main text. The Pt-SAs/MoS₂ catalyst delivers an extremely high exchange current density of 2.24 mA cm⁻², which is largely increased by a factor of 1.6 compare to commercial Pt/C;
- (3) Owing to 100% atom-utilization efficiency and unique electronic states, single-atom Pt usually shows superior HER activity compared to bulk Pt catalyst. (detailed explanation shown below).

Fig. 4 (a) HER polarization curves of bare, ce-MoS₂-, commercial Pt/C-, and Pt-SAs/MoS₂-covered GC electrodes in 0.5 M H₂SO₄ solution. Inset: the mass activity of Pt-SAs/MoS₂ normalized to the Pt loading at an overpotential of 50 mV in comparison with commercial Pt/C. (b) Tafel plots derived from the corresponding polarization curves.

Many previously reported works showed that the acidic HER activities of some single-atom Pt catalysts are much better than Pt nanoparticles [for example, *Nat. Catal.* 2018, 1, 985; *Nat. Energy* 2019, 4, 512; *Nat. Commun.* 2019, 10, 1657; *Angew. Chem.* 2018, 130, 9526; *Nat. Commun.* 2016, 7, 13638; *Nat. Commun.* 2020, 11, 1215; *Nat. Commun.* 2017, 8, 1490; *Sci. Adv.* 2018, 4, 6657; *Nat. Commun.* 2019, 10, 4936]. As the supported metals are downsized to the minimum, SACs reach the utmost atom-utilization efficiency. Moreover, the atomically dispersed metal species usually exhibit unique electronic states due to distinctive coordinated environments, strong metal-support interactions, and quantum size effects. The strong metal-support interactions, usually associated with charge transfer process, modulate the *d*-band structure of single-atom metal, strengthen the absorption of reaction intermediates, and hence lower the energy barrier and facilitate the rate-limiting step (Fig. R5a).

In our work, we focus on the big picture of site-specific electrodeposition methodology for the self-terminating growth of single-atom metals. And we further utilize Pt-SAs/MoS₂ to catalyse acidic HER as a model system, which shows the potential

application of the single-atom metals in heterogenous catalysis. Although a systematic experimental kinetic investigation is beyond the scope of the current work, the likely acidic HER mechanism of hydrogen recombination and desorption for single-atom Pt on the ce-MoS₂ is proposed using DFT calculations, with an attempt to obtain insights into the kinetics of HER. Actually, we are now trying to use some *in-situ* spectroscopies (NAP-XPS, XAS) to further elucidate the HER enhancement mechanism of single-atom Pt, which is still at the early stage, but perhaps can be used to tell why these catalysts are better than Pt nanoparticles. Now we only find that the absorption of H on single-atom Pt is perhaps different from that on Pt nanoparticle (example shown in Fig. R5, unpublished).

Fig. R5 (a) Schematic illustration of charged single-atom Pt for accelerating HER. (b) Schematic illustration of the combined synchrotron NAP-XPS with hydrogen absorption measurements for single-atom metal samples. (c) NAP-XPS spectrum of Pt 4f in single-atom metal samples under specific conditions.

2. Is it reasonable to be able to see single Pt atoms by HAADF-STEM, as suggested in Fig. 2c?

Response: We thank the reviewer for this question. It is reasonable to be able to see single Pt atoms by HAADF-STEM (as shown in Fig. 2c). HAADF-STEM is the most

convincing and intuitive approach to confirm the existence of isolated single metal atoms, which can directly image the single metal atoms, identify their location, and determine the spatial distribution [*Accounts Chem. Res.* 2013, 46, 1740; *Chem. Rev.* 2018, 118, 4981].

HAADF is an STEM technique which produces an annular dark field image formed by very high angle, incoherently scattered electrons. This technique is highly sensitive to variations in the atomic number of atoms in the sample (*Z*-contrast images). Image resolution in HAADF-STEM is very high and predominately determined by the size of the electron probe, which in turn depends on the ability to correct the aberrations of the objective lens, in particular the spherical aberration. One primary advantage of HAADF-STEM is that the imaging is based on Rutherford scattering in which image intensity for given atoms is roughly proportional to the square of the atomic number (Z^2) of the element, **allowing heavy metal atoms to brightly contrast against low background supports**. In brief, the higher the *Z* values of the elements, the brighter the images are shown. For example, the *Z*-contrast image characteristics of HAADF allows Pt atoms (*Z* = 78) to stand out as brighter spots above the MoS₂ support (Mo, *Z* = 42; S, *Z* = 16).

Single Pt atoms were also characterized by HAADF-STEM in the following literatures (Fig. R6) [*Nat. Nanotechnol.* 2018, 13, 411; *Nat. Catal.* 2018, 1, 985; *Nat. Energy* 2019, 4, 512; *Nat. Commun.* 2019, 10, 1657; *Angew. Chem.* 2018, 130, 9526; *Nat. Commun.* 2016, 7, 13638; *Nat. Commun.* 2020, 11, 1215; *Nat. Commun.* 2017, 8, 1490; *Sci. Adv.* 2018, 4, 6657; *Nat. Commun.* 2019, 10, 4936; *Angew. Chem.* 2019, 131, 1175].

Fig. R6 (a~l) HAADF-STEM images of various previously reported single-atom Pt catalysts.

3. For Fig. 2i, I believe that the x-axis should be "Radial distance" not "Radical distance".

Response: We really thank the reviewer for this comment and are sorry for the mistake. We have corrected the section in Fig. 2i, and further carefully check throughout the manuscript to avoid such mistakes.

4. Can you estimate the coverage of Cu in the UPD from the cyclic voltammograms? It is truly less than a monolayer?

Response: We thank the reviewer for this valuable comment and concern. We have given the answers as follows and added these explanations to the corresponding section (Supplementary Note 2) in Supplementary Materials.

We have estimated the coverage of Cu in the UPD from the cyclic voltammogram (see details in Supplementary Note 2). The amount of Cu can be obtained by integration of

the cathodic peak corresponding to Cu UPD ($2.156 \times 10^{-10} \text{ mol}$). The real coverage of Cu (2.76%) on the ce-MoS₂ nanosheets is far less than the ideal coverage (33.3%) based on the Cu-S pair ratio, which might originate from the existence of many S defects and localized structural distortion. However, this result indirectly confirms that SSED Cu on ce-MoS₂ is truly far less than a monolayer.

We further confirmed the successful SSED growth of single-atom Cu on ce-MoS₂ nanosheets by using TEM and XPS. No obvious nanoparticles or clusters were discernable in Cu-SAs/MoS₂ (TEM image, Supplementary Fig. 4g). The chemical configuration and binding status of Cu-SAs/MoS₂ were also characterized by XPS (Supplementary Fig. 4h). The main Cu $2p_{3/2}$ peak of Cu-SAs/MoS₂ at 932.9 eV is located between Cu⁰ (932.4 eV) and Cu²⁺ (934.7 eV), indicative of Cu species with partially positive charge owing to the electronic interaction between single Cu atom and ce-MoS₂. Importantly, the absence of Cu $2p$ satellite peaks (typically at *ca.* 940–944 eV) further confirms the ionic nature of Cu species in Cu-SAs/MoS₂. Simultaneously, the average binding energy of S $2p$ decreases and its shape becomes slightly more pronounced after functionalization of Cu atoms compared to that of pure ce-MoS₂, which in turn confirms the attachment of Cu atoms onto the S atoms.

Additionally, owing to a local one-to-one galvanic exchange of Cu atoms, the existence of single-atom Pt can be used to indirectly confirm that Cu synthesized by site-specific UPD on ce-MoS₂ was truly atomically dispersed instead of a monolayer. In order to systematically characterize single-atom Pt, various techniques have been used, such as HAADF-STEM, EXAFS, XANES, XPS, ICP and DFT simulations.

5. The CVs are not as sharp as usually observed for UPD. Is there an explanation for this? Would it be possible to plot the CVs using current density instead of current?

Response: We thank the reviewer for this comment. We have plotted the CVs using current density instead of current (see Supplementary Fig. 4). The explanation of broad CVs for UPD and some examples are given below.

Underpotential deposition (UPD) is a surface-dominated process. The shapes, positions, and number of the UPD peaks depend on the substrate and the crystal plane on which the adsorption takes place, as well as on the nature of the electrolyte [*Surf. Sci.* 1976, 54, 489; *Electrochim. Acta* 1976, 21, 967; *Sci. Technol. B* 1991, 9, 969; *Phys. Chem.* 1987, 91, 3494]. On a single-crystal metal electrode, a very dense monolayered metal film can be formed *via* underpotential deposition. Such atom-by-atom metal film structure leads to the strong mutual interaction between metal atoms. In that case, the energy level of each deposited metal tends to be equal, and thus a very sharp current peak can be achieved [*J. Chem. Phys.* 1996, 104, 5699]. **In our system, the underpotentially deposited single-atom metals are totally isolated and show no interactions with each other. Owing to the existence of nonuniformly distributed defects and localized structural distortion on substrate, energy level of each single-atom metal becomes a little distinctive, leading to the CV broadening.** Similar phenomena were also shown in many works (Fig. R7a) [*Nat. Mater.* **12**, 850 (2013); *Nat. Catal.* **1**, 460-468 (2018); *Nature* **332**, 426-429 (1988)].

In principle, the anodic and cathodic currents should be symmetric about the potential axis in the CVs. For instance, current-potential data for Cu UPD on single-crystal Au(111) surfaces, obtained from cyclic voltammetry CV experiments (Fig. R7b), display two well-defined, symmetric and sharp pairs of peaks, corresponding to $(\sqrt{3} \times \sqrt{3})\mathcal{R}30^\circ$ honeycomb structure and (1×1) structure [*J. Electroanal. Chem.* 1991, 315, 275]. Such formation of sharp peaks is accompanied by simultaneous and rapid deposition of large amounts of Cu atoms. Note that the deposition peak at low underpotentials splits in two on high-quality Au(111) single crystals, which arises from two different nucleation processes, one taking place on surface defects and the other on well-ordered (111) terraces. Different from that on the single-crystal metal electrode,

current-potential data for Pb UPD on Au polycrystalline and different kinds of AuNPs show much broader and less symmetric peaks (Fig. R7c and R7d) [*Surfaces* 2019, 2, 257]. The CV broadening can be ascribed to the existence of various facets consisting of different terraces, ledges, and kinks.

Fig. R7 (a) CV measurement of the Li-PPS nanosheets-modified GCE (red line) in N_2 -saturated 0.1 M H_2SO_4 solution containing 2 mM $CuSO_4$ [*Nat. Catal.* **1**, 460-468 (2018)]. (b) CV curve for Au(111) single-crystal electrode in 0.05 M H_2SO_4 solution containing 1 mM $CuSO_4$ [*J. Electroanal. Chem.* 1991, 315, 275]. (c) CV curve of Au polycrystalline 0.1 M NaOH + 1 mM $Pb(NO_3)_2$ solution [*Surfaces* 2019, 2, 257]. (d) CV curves of different AuNPs (nanospheres AuNSs, nanorods AuNRs, and nanocubes AuNCs) in 0.1 M NaOH + 1 mM $Pb(NO_3)_2$ solution [*Surfaces* 2019, 2, 257].

We look forward to your positive reply.

Best regards

Dr. Xing-Hua Xia (on behalf of all authors)

Professor

School of Chemistry and Chemical Engineering

Nanjing University

E-mail: xhxia@nju.edu.cn

Reviewers' Comments:

Reviewer #1:

Remarks to the Author:

My previous comments have been addressed and the revised manuscript can be accepted now.

Reviewer #2:

Remarks to the Author:

The authors have done very good work of improving the manuscript. My and the other referees comments have been taken into account and the manuscript and the supplementary have improved substantially. I am satisfied to the modifications and in my opinion the manuscript can now be published as it is.

Reviewer #3:

Remarks to the Author:

I am satisfied with the authors' revision of the manuscript. They have addressed the issues that I raised in my initial review. I think that the addition of exchange current densities and the comparison with bulk Pt for the HER reaction is a good addition. I also think it was a good addition to make it clear how this work differed from previous single-atom catalysis work. I think it is now ready for publication.

REVIEWERS' COMMENTS:

Reviewer #1 (Remarks to the Author):

My previous comments have been addressed and the revised manuscript can be accepted now.

Response: We really thank the reviewer for his/her positive comments.

Reviewer #2 (Remarks to the Author):

The authors have done very good work of improving the manuscript. My and the other referees comments have been taken into account and the manuscript and the supplementary have improved substantially. I am satisfied to the modifications and in my opinion the manuscript can now be published as it is.

Response: We really thank the reviewer for his/her positive comments.

Reviewer #3 (Remarks to the Author):

I am satisfied with the authors' revision of the manuscript. They have addressed the issues that I raised in my initial review. I think that the addition of exchange current densities and the comparison with bulk Pt for the HER reaction is a good addition. I also think it was a good addition to make it clear how this work differed from previous single-atom catalysis work. I think it is now ready for publication.

Response: We really thank the reviewer for his/her positive comments.